# Polyvinylpyrrolidone Nanofibers Encapsulating an Anhydrous Preparation of Fluorescent SiO_2_–Tb^3+^ Nanoparticles

**DOI:** 10.3390/nano9040510

**Published:** 2019-04-02

**Authors:** Jianhang Shi, Yanxin Wang, Linjun Huang, Peng Lu, Qiuyu Sun, Yao Wang, Jianguo Tang, Laurence A. Belfiore, Matt J. Kipper

**Affiliations:** 1Institute of Hybrid Materials, National Center of International Joint Research for Hybrid Materials Technology, National Base of International Sci. & Tech. Cooperation on Hybrid Materials, College of Materials Science and Engineering, Qingdao University, 308 Ningxia Road, Qingdao 266071, China; sjhang@163.com (J.S.); newboy66@126.com (L.H.); 18753360989@163.com (P.L.); SQY17861431023@163.com (Q.S.); wangyaoqdu@126.com (Y.W.); 2College of Materials Science and Engineering, Qingdao University, Qingdao 266071, China; 3Department of Chemical and Biological Engineering, Colorado State University, Fort Collins, CO 80523, USA; laurence.belfiore@colostate.edu (L.A.B.); matthew.kipper@colostate.edu (M.J.K.); 4School of Biomedical Engineering, Colorado State University, Fort Collins, CO 80523, USA; 5School of Advanced Materials Discovery, Colorado State University, Fort Collins, CO 80523, USA

**Keywords:** terbium complex, silica, electrospinning, fluorescence, lifetime

## Abstract

A novel anhydrous preparation of silica (SiO_2_)-encapsulated terbium (Tb^3+^) complex nanoparticles has been investigated. The SiO_2_-Tb^3+^ nanoparticles are incorporated in electrospun polyvinylpyrrolidone hybrid nanofibers. Transmission electron microscopy confirms that Tb^3+^ complexes are uniformly and stably encapsulated in or carried by nanosilica. The influence of pH on the fluorescence of Tb^3+^ complexes is discussed. The properties, composition, structure, and luminescence of the resulting SiO_2_–Tb^3+^ hybrid nanoparticles are investigated in detail. There is an increase in the fluorescence lifetime of SiO_2_–Tb^3+^ nanoparticles and SiO_2_–Tb^3+^/polyvinylpyrrolidone (PVP) hybrid nanofibers compared with the pure Tb^3+^ complexes. Due to the enhanced optical properties, the fluorescent hybrid nanofibers have potential applications as photonic and photoluminescent materials.

## 1. Introduction

Studies on rare-earth-doped polymer-based light-emitting materials are of great interest, because they have played a pivotal role in the fields of medical diagnostics, displays, and detection systems [1]. Although the organic light emitters based on polymer materials show very good prospects, unfortunately, these materials have limited color purity and low luminous efficiency [2,3]. Therefore, finding materials with stable, narrow-linewidth, and high luminous efficiency is the purpose of much research. Rare-earth elements have a special electron configuration, imparting uncommon photonic properties. The luminescence characteristics are mainly based on the transitions of their 4f electrons within the f–f configuration or between the f–d configurations. Since the f–f transitions of rare-earth ions belong to forbidden transitions, the absorption cross section of rare-earth ions in the visible or ultraviolet region is very small. However, organic ligands often have large absorption cross sections in the ultraviolet region, and energy can be transferred to rare-earth ions through intramolecular energy transfer, greatly improving the emission intensity of rare-earth ions [4,5,6]. Rino et al. successfully prepared a Tb(acac)_3_phen complex and used it as an emissive layer in organic light Emitting diodes (OLED) [2]. Zheng developed Tb(acac)_3_AAP and worked on its photoluminescent and electroluminescent properties [3]. Bukvetskii studied the atomic structure of crystals of Tb(acac)_3_phen, which were characterized by means of the X-ray structural analysis method [7]. We have selected terbium (Tb^3+^) elements with strong luminescent properties (characteristic green fluorescence) as a luminous center. In addition, the matching of energy levels between rare-earth ions and organic ligands is one of the main factors affecting the luminescent efficiency of rare-earth complexes; because of the energy levels of terbium and acetylacetone (acac), they are matched properly. In this section, acetylacetone and phenanthroline are used as ligands to synthesize Tb (acac)_3_phen complexes.

Rare-earth ion complexes with organic ligands have many advantages compared to other fluorophore types, such as narrow emission bands, high quantum yields, and nontoxicity. However, rare-earth complexes are unstable at high temperatures and in the presence of acids. In a complex environment, the quantum yield of the rare-earth complexes may decrease. SiO_2_ is a versatile material that has exceptional compatibility in many biomedical applications, with its optical transparency, chemical inertness, and easy generation. SiO_2_ can protect rare-earth complexes, improving their thermal stability for the preparation of durable, functional materials [7]. Tagaya et al. successfully prepared Eu(III)-doped nanoporous silica spheres by the sol–gel method and found that Eu^3+^ ions were located in a low-symmetry environment [8]. Mukhametshina studied the effects of silica coating and further silica surface decoration by phospholipid bilayers on the quenching of Tb(III) complexes by adrenochrome [9]. Binnemans provided detailed information on certain materials, including sol–gel technology-based composites using silica precursors as the host matrix [10].

Herein, we report a new route for anhydrous preparation of SiO_2_ nanoparticles containing fluorescent Tb^3+^ complexes with acetyl acetone (acac) and phenanthroline (phen) (SiO_2_–Tb^3+^). We also report the effect of pH on the fluorescence of Tb^3+^ complexes. To obtain stable fluorescent hybrid nanospheres, an anhydrous preparation was used to prepare spherical SiO_2_ doped with Tb(acac)_3_phen complexes, resulting in uniform particle size. Polyvinylpyrrolidone (PVP) ultrafine fibers containing SiO_2_–Tb^3+^ nanoparticles were prepared by electrospinning. This new method for the preparation of fluorescent hybrid nanofibers has potential applications in the field of organic light-emitting materials.

## 2. Materials and Methods

### 2.1. Materials

Terbium oxide (Tb_4_O_7_, 99.99%, A.R.), 1,10-phenanthroline (Phen, 99%, A.R.), acetylacetone (Hacac, 98%, A.R.), hydrochloric acid (HCL, 38%, A.R.), hydrogen peroxide (H_2_O_2_, 30%, A.R.), sodium hydroxide (NaOH, A.R.), tetraethyl orthosilicate (TEOS, 99 wt. %, A.R.), ammonium hydroxide (NH_3_∙H_2_O, 28%, A.R.), and polyvinylpyrrolidone (PVP, ${\bar{M}}_{W}$ = 1,300,000) were analytically pure and purchased from China National Medicines Group (Beijing, China).

### 2.2. Characterization

A JEOL JEM-2100F (JEOL Inc., Tokyo, Japan) transmission electron microscope was used for the identification of morphology and size of hybrid nanoparticles. The structure and crystal phases of Tb(acac)_3_phen and SiO_2_–Tb^3+^ hybrid nanofibers were determined by powder X-ray diffraction (XRD, Ultima IV, Rigaku Corporation, Tokyo, Japan). High-resolution transmission electron microscopy (HRTEM) was performed using a FEI Talos F200i microscope (Thermo Fisher Scientific Inc., Waltham, MA, USA) operating at 200 kV. The elemental composition was determined using scanning transmission electron microscopy with energy-dispersive X-ray spectroscopy (STEM-EDS) using a FEI Tecnai G2 F20 S-TWIN (FEI Inc., Hillsboro, OR, USA). Scanning electron microscope (SEM) images of the electrospun fibers were obtained using a SIGMA 500/PV (SIGMA Inc., St. Louis, MO, USA), with the electron microscope operating at 200 kV. The luminescence spectra and lifetime measurements were performed using an Edinburgh FLS1000 (Edinburgh Inc., Livingston, UK) photoluminescence spectrometer.

### 2.3. Preparation of Terbium Complexes with Different pH

Preparation of Tb(acac)_3_phen complex: 0.5 g Tb_4_O_7_ was added to 10 mL of H_2_O_2_ and then stirred at room temperature for 0.5 h. Then, 4 mL of hydrochloric acid (HCL, 38%) was added to the solution and the reaction mixture was stirred until a completely transparent solution had formed. The solution was crystallized at 60 °C and dried at 50 °C in the oven for more than 12 h, to obtain white crystallized powder of terbium chloride hexahydrate. Anhydrous ethanol was added to some of the powder to make a 0.1 mol/L TbCl_3_ solution, marked solution A. Acetylacetone (0.601 g, 6 mmol) and 1,10-phenanthroline (0.360 g, 2 mmol) were added to a flask, followed by addition of anhydrous ethanol under stirring to obtain 20 mL total volume, marked solution B. Then, 20 mL 0.1 mol/L solution A (2 mmol) was added dropwise to solution B while stirring, and the mixture was allowed to react for 2 h, to obtain Tb(acac)_3_phen complexes in solution.

The resulting 40 mL Tb(acac)_3_phen solution was divided into eight conical flasks, and the pH of each of these solutions containing the complexes was adjusted to either 6, 7, 8, 9, 10, 11, 12, or 13 by adding 0.1 mol/L sodium hydroxide (NaOH) solution. These eight Tb(acac)_3_phen solutions were centrifuged (speed 10,000 r/min, 20 min) and washed with ethanol two times, and then 5 mL of ethanol was added dropwise to form a stable and clear solution. These solutions were marked solutions C_1_–C_8_.

### 2.4. Anhydrous Preparation of SiO_2_–Tb^3+^ Nanoparticles

Three milliliters of NH_3_∙H_2_O was added to 50 mL of anhydrous ethanol; the mixed solution was vigorously stirred for 20 min to promote hydrolysis and to adjust the pH. Then, 1.5 mL tetraethoxysilane (TEOS) was added dropwise to the above solution. The mixed solution was vigorously stirred for 24 h at room temperature. The resulting solution was marked solution D. Two milliliters of solution D were added separately to solutions C_1_ through C_8_, and the mixed solutions were vigorously stirred for 4 h, centrifuged (speed 10,000 r/min, 20 min), washed with ethanol two times, and dried in an oven at 50 °C to form a white crystallized powder, which was the SiO_2_/Tb(acac)_3_phen (SiO_2_–Tb^3+^) hybrid nanoparticles.

### 2.5. Preparation of PVP-Based Fluorescence Solution

The SiO_2_–Tb^3+^ nanoparticles with a concentration of 0.01 mol/L were added to an anhydrous mixture of ethanol and N,N-Dimethylformamide (DMF) (1:1 *w*/*w*) at room temperature under continuous stirring, and subsequently, was stirred for 2 h at room temperature. PVP electrospinning solution was prepared by dissolving 1.739 g PVP into 20 g of the above solution, which was stirred for about 10 h, to ensure complete dissolution of the PVP and to form a uniform SiO_2_–Tb^3+^/PVP electrospinning solution.

### 2.6. Electrospinning of PVP-Based Fluorescence Nanofibers

The resulting stable homogeneous solution was loaded into a 5-mL plastic syringe for electrospinning at the same temperature (20 °C) and with a tip-to-collector distance of 20 cm. The needle (22 G) was connected to a high-voltage power source and the SiO_2_–Tb^3+^/PVP solution was fed at a constant rate (0.033 mL/min). A piece of flat aluminum foil was placed on the receiving board to serve as the collection electrode. The nanofibers were spun at a voltage of 29 kV. Figure 1 is a schematic showing the structure of the resulting fibers.

## 3. Results and Discussion

The morphological and structural feature of the resulting samples were examined by TEM; from this, it is clear that SiO_2_ nanoparticles are uniform and nonagglomerated with monodisperse spheres with average diameter of about 120 nm, as shown in Figure 2a. SiO_2_ and Tb(acac)_3_phen form composite nanospheres, and then SiO_2_–Tb^3+^ nanospheres still maintain good dispersion and the same size. From Figure 2a,b, we can see that most of the Tb^3+^ complexes are attached to the SiO_2_ surfaces and a small amount of them are embedded inside the SiO_2_ nanoparticles. Figure 2d–h shows the dark-field scanning transmission electron microscopy (STEM) images and the elemental mapping of C (red), N (white), O (indigo blue), Si (yellow), and Tb (green). Detection of C, Tb, and N confirms the presence of the Tb(acac)_3_phen complexes in the SiO_2_ nanoparticles.

Figure 3a shows fluorescence properties of Tb(acac)_3_phen complexes at different neutral or alkaline pH (pH = 6, 7, 8, 9 10, 11, 12, or 13). Understanding the influence of pH may also give insight into the influence that the environment has on the overall functionality of the complexes. The fluorescence spectrum of each solution was recorded upon incremental addition of base (NaOH) at an excitation wavelength of 325 nm. A significant increase of Tb(acac)_3_phen emission intensity occurs as the pH is raised from pH = 6 to pH = 9, while an obvious reduction in Tb(acac)_3_phen emission is observed from pH = 9 to pH = 13. Under the condition of strong acid, the fluorescence intensity of Tb(acac)_3_phen drops sharply, due in part to the protonation of the imines, which reduces the complexation with Tb^3+^ [11]. In the strong alkali solution, Tb^3+^ and OH^−^ produce Tb(OH)_3_ precipitation, also reducing the concentration of the Tb(acac)_3_phen complex. To further investigate the luminescence properties of Tb(acac)_3_phen complexes at different pH, the room-temperature luminescence decay curves are presented in Figure 3b: excited at 325 nm and monitored at the ^5^D_4_–^7^F_5_ emission line. The decay curves fit a single exponential function: ${D(t)=c}_{0}\exp(-t/\tau )$ [12]. The emission lifetimes of Tb(acac)_3_phen complexes are ranged from 0.36 ms to 1.28 ms (Table 1). In fluorescence spectroscopy, the quantum yield and fluorescence lifetime are determined by the decay from radiative, Γ, and nonradiative, *k_nr_*, mechanisms. According to the Jablonski energy level table [13], for free fluorescent molecules without other special quenching processes, the quantum yield, Q, and fluorescence lifetime, *τ*_0_, are:(1)$$Q_{0}=\frac{\Gamma }{\Gamma +k_{nr}}\;\tau_{0}={\left(\Gamma +k_{nr}\right)}^{-1}$$

It can be seen from Equation (1) that the greater the radiation attenuation, the higher the quantum yield of the fluorescent molecule and the shorter the lifetime. The radiation attenuation for the terbium ion is a fixed constant. On the other hand, NaOH is used to adjust the solution pH in our experiment. Therefore, OH^−^ ions are the major origin of fluorescence efficiency change. Since samples prepared at different pH have the same apparent Tb(acac)_3_phen crystal structure, OH^−^ ions are apparently behaving as efficient quenchers of the luminescence of Tb^3+^ through multiphonon relaxation above pH 9. In samples of Tb(acac)_3_phen complexes from pH 6 to 9, the concentration of OH^−^ ions is orders of magnitude lower, resulting in increased lifetimes compared to the conditions at higher alkalinity. Therefore, remarkably increasing fluorescence efficiency and lifetime are observed in samples of Tb(acac)_3_phen complexes from pH 6 to 9. It is seen from Figure 3b that the fluorescence decay curve of Tb(acac)_3_phen complexes is nearly linear on the semilog plot, indicating almost single-exponential fluorescence decay at the lower pH conditions. The nonradiative decay of ^5^D_4_ of Tb^3+^ should include the nonradiative transition of ^5^D_4_–^7^F_j_ (j = 3, 4, 5, 6) and the energy transfer from ^5^D_4_ to other Tb^3+^ ions. The above results indicate that the nonradiative transition caused by OH^−^ ions is the leading contribution to the nonradiative decay processes of the ^5^D_4_ of Tb^3+^, and that the energy transfer from ^5^D_4_ of a Tb^3+^ to other Tb^3+^ ions may be negligible [11,14,15]. Therefore, Figure 3a,b indicates that tunable fluorescent lifetimes with various intensities are achieved by changing the pH of Tb(acac)_3_phen complexes, which can be potentially useful for multichannel bioimaging and optoelectronics [16].

The normalized excitation and emission spectra for the Tb(acac)_3_phen complex and SiO_2_–Tb^3+^ hybrid nanoparticles are shown in Figure 4a,b, respectively. The excitation spectra of Tb(acac)_3_phen shows a wide band, which is mainly attributed to the π–π* electron transfer of ligands. The excitation spectrum for the Tb(acac)_3_phen complex is very different from that of the SiO_2_–Tb^3+^ hybrid nanoparticles. In the excitation spectra of SiO_2_–Tb^3+^ nanospheres, the corresponding broadband absorption transition of organic ligand b–diketone still exists. The line width of the excitation spectrum is 25 nm, but the absorption band position of organic ligand remains unchanged; the excitation spectrum of SiO_2_–Tb^3+^ nanospheres is narrower compared with the pure Tb(acac)_3_phen complex, because intramolecular energy transfer occurs much more effectively between the ligands and Tb^3+^ following silica encapsulation. The excitation peaks are assigned to the π–π* transition of the acetylacetone and phenanthroline [17]. The absorption at 325 nm in the excitation spectrum of SiO_2_–Tb^3+^ is stronger than the corresponding absorption in the Tb(acac)_3_phen complex, indicating that SiO_2_ nanoparticles participate in the coordination of phenanthroline and acetylacetone molecules. An anhydrous preparation of SiO_2_–Tb^3+^ nanoparticles leads to different symmetry of the environment. Due to the effect of the rigid silica shell, the nanoclusters of rare-earth complexes cannot be agglomerated. The specific surface area of rare-earth complex nanoclusters is larger, and there are more molecules on the surface than those occupying the interior of clusters. We consider that the increase of excitation intensity is due to the increase of molecules on the surface of rare-earth complexes. The strongest fluorescence is obtained by excitation at 325 nm, indicating that the fluorescence intensity of the complex is strongest under this excitation wavelength. Both the emission spectra of Tb(acac)_3_phen and SiO_2_–Tb^3+^ are obtained by monitoring the strongest emission wavelength of the Tb^3+^ ions at 548 nm. As can be seen from Figure 4b, the characteristic emission peaks of Tb^3+^ appear at 491 nm, 548 nm, 581 nm, and 620 nm. These bands are attributed to the ^5^D_4_→^7^F_6_, ^5^D_4_→^7^F_5_, ^5^D_4_→^7^F_4_, and ^5^D_4_→^7^F_3_ transitions of Tb^3+^ ions. The ^5^D_4_→^7^F_5_ transition at the emission wavelength of 548 nm is the hypersensitive transition with strongest fluorescence intensity [17,18,19]. The SiO_2_–Tb^3+^ hybrid nanoparticles have stronger luminescence than the corresponding pure Tb(acac)_3_phen complex; the electric dipole transition and its corresponding intense emission at 548 nm aree significantly increased [20]. The amorphous silica nanoparticles provide a microenvironment for the terbium molecules that decrease nonradiative transitions by confinement, resulting in the increasing of fluorescence intensity of SiO_2_–Tb^3+^ [21]. Figure 4c displays the FT-IR spectra of the Tb(acac)_3_phen complex and SiO_2_–Tb^3+^ hybrid nanoparticles. The vibrations located at the wavenumbers of 1046 cm^−1^ and 1086 cm^−1^ observed in the SiO_2_–Tb^3+^ spectrum are evidence of coordination bonds between the Tb(acac)_3_phen complex and SiO_2_ nanoparticles [22]. The Tb(acac)_3_phen complex powder shows a sharp and strong crystalline peak at the diffraction angle of 2θ = 10.36 (Figure 4d). Strong crystalline peaks are also observed at 2θ = 9.360, 11.460, 12.420, and 13.460, indicating that both Tb(acac)_3_phen and SiO_2_–Tb^3+^ hybrid nanoparticles are highly crystalline and that the crystals are relatively regular [18,23]. The Tb(acac)_3_phen complex and SiO_2_–Tb^3+^ hybrid nanoparticles have the same crystalline peak positions, indicating that the encapsulation of Tb(acac)_3_phen complexes in silica nanoparticles does not change the crystal structure of the Tb(acac)_3_phen complex.

The SiO_2_–Tb^3+^/PVP fluorescent hybrid materials were analyzed by HRTEM, SEM, and photoluminescence spectrometry. The SiO_2_–Tb^3+^/PVP hybrid nanofibers display bright-green light upon radiation with ultraviolet light at 365 nm. To comprehensively reveal the underlying distribution of nano-SiO_2_–Tb^3+^ in PVP fibers, the elemental distribution of the SiO_2_–Tb^3+^/PVP hybrid nanofibers was evaluated with scanning transmission electron microscopy (STEM) (Figure 5a). From Figure 5b, it can be seen that the SiO_2_–Tb^3+^ particles were not affected by electrospinning, as the structure of the SiO_2_–Tb^3+^ nanospheres is well preserved.

Figure 5d–f shows the elemental mapping analysis of C (red), O (violet), and Si (indigo-blue), demonstrating that SiO_2_–Tb^3+^ hybrid nanoparticles are successfully encapsulated in the PVP nanofibers. From Figure 5a, it can be seen that SiO_2_–Tb^3+^ hybrid nanoparticles are well separated and are compatibilized with the polymer [4].

Figure 6a shows the excitation at the absorption band (325 nm) of the SiO_2_–Tb^3+^/PVP fibers. The emission spectrum of the SiO_2_–Tb^3+^/PVP fibers is red-shifted compared with the fluorescence spectrum of the SiO_2_–Tb^3+^ hybrid nanoparticles. This is evidence that although the Tb^3+^ complex is prepared so as to have complete coordination, the complexes are still able to interact with the PVP. SiO_2_–Tb^3+^/PVP nanofibers show strong green luminescence and exhibit Tb^3+^ corresponding to the characteristic transition of ^5^D_4_–^7^F_J_ (J = 6, 5, 4, 3). The green emission at 548 nm from the electronic dipole transition ^5^D_4_–^7^F_5_ possesses the strongest intensity, indicating that the position of Tb^3+^ is in an environment without reverse symmetry [24]. Strong ^5^D_4_–^7^F_5_ peak intensity indicates a highly polarized chemical environment around the Tb^3+^ ion and is responsible for producing bright-green light upon 325-nm ultraviolet radiation. The emission lifetimes of SiO_2_–Tb^3+^ hybrid nanoparticles and SiO_2_–Tb^3+^/PVP nanofibers are 1.54 ms and 1.83 ms, respectively, and are shown in Figure 6b. SiO_2_–Tb^3+^/PVP hybrid fibers have longer fluorescence lifetimes compared with SiO_2_–Tb^3+^ nanoparticles. The main factor affecting the fluorescence lifetimes (τ) are the radiative transition rate (A_rad_) and nonradiative transition rate (A_nrad_) of Tb^3+^ ions. The relationship between A_rad_ and A_nrad_ is $\tau =1/(A_{\mathrm{rad}}+A_{\mathrm{nrad}})$ [25]. TEM shows that SiO_2_–Tb^3+^ nanoparticles are enveloped by PVP polymer, which demonstrates that the site of Tb^3+^ ions is located in an environment without inversion symmetry, and effectively increasing the rate of radiative transition (A_rad_). The change of radiation transition rate is dependent on the index of refraction on the border molecule/polymer. The increase of radiation transition rate to Tb^3+^ in SiO_2_–Tb^3+^/PVP hybrid fibers is due to the increase of the refractive index of the external medium. At the same time, due to the relatively regular dispersion of SiO_2_–Tb^3+^ nanoparticles in PVP nanofibers, the rate of nonradiative transition (A_nrad_) decreases with the decrease of energy transfer between molecules. Nonradiative transitions between rare-earth ions can transfer energy through phonons. Rare-earth ions emit multiple phonons from the excited state at the same time, then relaxing to the next adjacent level. This process is called multiphonon nonradiative relaxation. The nonradiative decay of ^5^D_4_ of Tb^3+^ includes the nonradiative transition of ^5^D_4_–^7^F_j_ (j = 3, 4, 5, 6) and the energy transfer from ^5^D_4_ to other Tb^3+^ ions. Given the relatively regular dispersion of SiO_2_–Tb^3+^ nanoparticles in PVP nanofibers, the energy shift caused by energy transfer of phonons can be neglected. The surface defect states (OH molecules) of SiO_2_ nanoparticles decrease sharply after SiO_2_–Tb^3+^ nanoparticles are doped into PVP fibers; while OH^−^ ions have larger vibration energy, they can greatly increase the multiphonon nonradiative relaxation between energy levels, and therefore the multiphonon relaxation between energy levels decreases after the reduction of OH^−^ ions, and then the nonradiative transition probability decreases. Therefore, when the reduction of the nonradiative transition rate (A_nrad_) is greater than the increase of the radiative transition rate (A_rad_), the sum of the two factors will decrease. As a result, the fluorescence lifetimes of the SiO_2_–Tb^3+^/PVP hybrid fibers is longer than that of the SiO_2_–Tb^3+^ nanoparticles, improving the luminescence monochromaticity. However, the longer the fluorescence lifetime is, the greater the energy storage capacity [26,27].

## 4. Conclusions

In conclusion, in order to obtain stable fluorescent hybrid nanospheres, silica nanoparticles were prepared which encapsulated Tb^3+^ complexes, using a novel hydrolysis method without water in the reaction. STEM combined with photoluminescence spectrometry confirms that inorganic–organic SiO_2_–Tb^3+^ hybrid nanoparticles with uniform morphology, stable properties, and long luminescence lifetimes were successfully synthesized by this method. The preparation method reported here results in regular silica nanoparticles with stable structures, and the preparation method is much simpler than the traditional method. Due to the improved photoluminescence properties of SiO_2_–Tb^3+^ hybrid nanoparticles encapsulated in PVP polymers, these materials may be introduced into OLED and luminescence probe applications. The results of this study provide new insight into the design of long-luminescence-lifetime materials.

## Figures and Tables

**Figure 1 nanomaterials-09-00510-f001:**
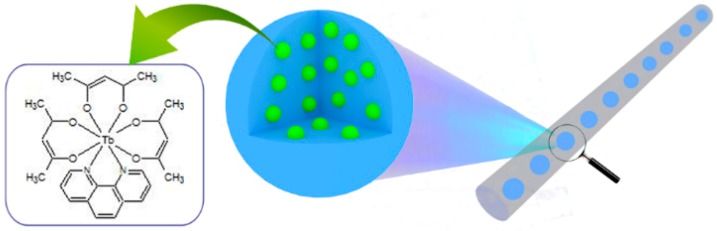
Schematic illustration of electrospun fiber containing luminescent SiO_2_–Tb^3+^ nanoparticles.

**Figure 2 nanomaterials-09-00510-f002:**
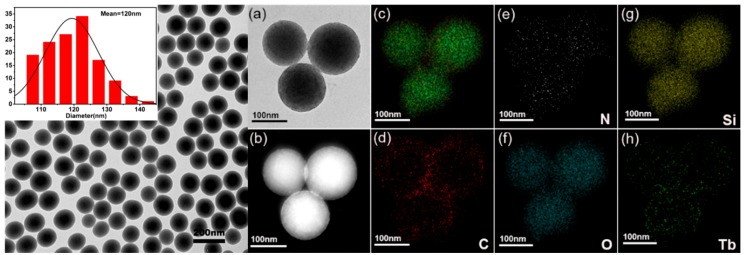
TEM images and the particle size distribution of the SiO_2_ nanoparticles; (**a**) TEM image of SiO_2_–Tb^3+^ hybrid nanoparticles; (**b**) STEM dark-field (DF) image; (**c**) merged element maps; (**d**–**h**) elemental mapping of C, N, O, Si, and Tb elements of SiO_2_–Tb^3+^ hybrid nanoparticles, individually.

**Figure 3 nanomaterials-09-00510-f003:**
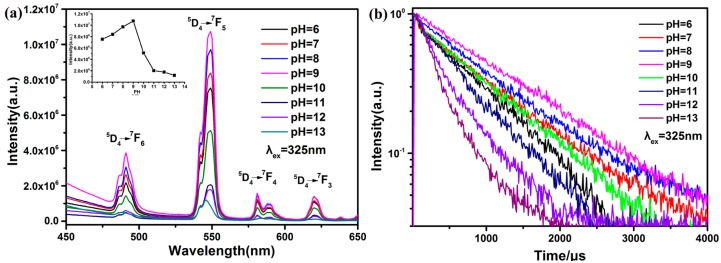
Changes in fluorescence emission and fluorescence lifetime for Tb(acac)_3_phen complexes as a function of pH; (**a**) Fluorescence spectra of Tb(acac)_3_phen complex (inset shows changes at 548 nm, λ_ex_ = 325 nm); (**b**) Room-temperature fluorescence decay curves of Tb(acac)_3_phen when excited at 325 nm and monitored at 548 nm.

**Figure 4 nanomaterials-09-00510-f004:**
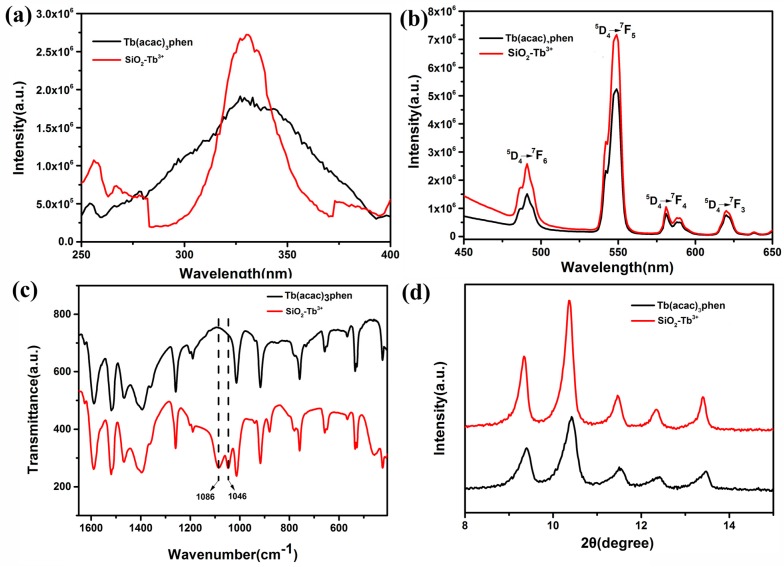
Excitation (λ_em_ = 548 nm) (**a**) and emission (λ_ex_ = 325 nm) (**b**) spectra of the Tb(acac)_3_phen complex and SiO_2_–Tb^3+^ hybrid nanoparticles; (**c**) FT-IR spectra of Tb(acac)_3_phen and SiO_2_–Tb^3+^; (**d**) X-ray diffraction patterns of the Tb(acac)_3_phen complex and SiO_2_–Tb^3+^ hybrid nanoparticles.

**Figure 5 nanomaterials-09-00510-f005:**
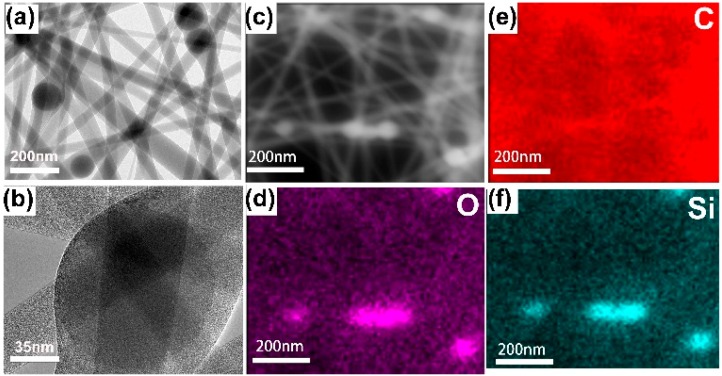
(**a**) TEM image of SiO_2_–Tb^3+^/PVP nanofibers; (**b**) HRTEM image of SiO_2_–Tb^3+^/PVP nanofibers; (**c**) SEM dark-field image of SiO_2_–Tb^3+^/PVP nanofibers; (**d**–**f**) Elemental mapping of O, C, Si, elements of SiO_2_–Tb^3+^ hybrid nanoparticles in SiO_2_–Tb^3+^/PVP nanofibers, individually.

**Figure 6 nanomaterials-09-00510-f006:**
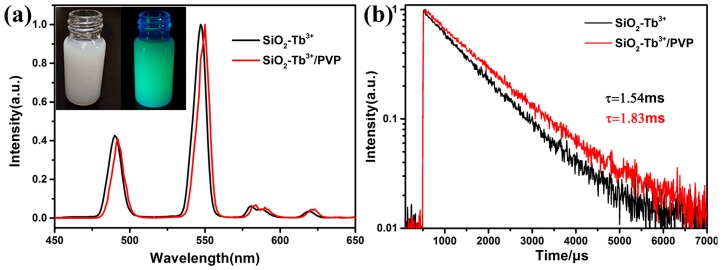
Emission (**a**) spectra and the ^5^D_4_–^7^F_5_ emission decay curves (λ_ex_ = 325 nm) (**b**) of SiO_2_–Tb^3+^ hybrid nanoparticles and SiO_2_–Tb^3+^/PVP nanofibers at 548 nm. The inset of the Figure 6a is SiO2@Tb(acac)3phen/PVP spinning solution under 365 ultraviolet light.

**Table 1 nanomaterials-09-00510-t001:** The emission lifetimes of Tb(acac)_3_phen complexes with different pH.

Samples	1	2	3	4	5	6	7	8
pH	6	7	8	9	10	11	12	13
τ/ms	0.75	0.88	1.04	1.28	0.82	0.75	0.44	0.36

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
