# Peer review of "Polyvinylpyrrolidone Nanofibers Encapsulating an Anhydrous Preparation of Fluorescent SiO2–Tb3+ Nanoparticles"

_nanomaterials, 2019, doi:10.3390/nano9040510_

Round 1
Reviewer 1 Report
The paper deal with luminescent properties of the SiO2-Tb3+/PVP composite. Several corrections and explanations should be done before considering for publication:
- In the materials source of phenanthroline, acetylacetone, phenanthroline should be added
- Authors explain lower emission intensity at lower pH as “Under the condition of strong acid, the fluorescence intensity of Tb(acac)3phen drops sharply, due in part to the protonatation of the imines, which reduces the complexation with Tb3+”, but preparation method include strong acid (HCl) for all samples, and in the next steps only sodium hydroxide is added. It means that pH only increases and is alkaline. Therefore this explanation cannot be applied here.
- In the figure 3a instead of energy diagram it would be better to add integrated emission intensity in the function of pH, because from the picture it is not clear how the emission intensity changes.
- Excitation wavelength should be added in the Fig 3a,b
- Authors wrote “by monitoring the strongest emission wavelength of the Tb3+ ions at 548 nm. These bands are attributed to the 5D4→7F6, 5D4→7F5, 5D4→7F4, and 5D4 →7F3 transitions of Tb3+ ions, respectively”, but emission at 548 is assigned only to , 5D4→7F5 transition
- Authors wrote “The excitation energy emits the strongest fluorescence at 325 nm.” or “The emission peaks are assigned to the π-π* transition of the acetylacetone and phenanthroline” In the excitation spectra at 325 we don’t observe emission. It is energy that is absorbed to emit at 548 nm. Therefore this sentence is not correct.
- In one sentence authors wrote “The SiO2-Tb3+/PVP hybrid nanofibers display bright green light upon radiation with ultraviolet light at 365 nm.” and in another “…the Tb3+ ion and is responsible for producing bright green light upon 325 nm ultraviolet radiation”. What is the real excitation wavelength used n the experiment. Another thing is that it should be also added in Fig 6.
- Authors wrote “At the same time, due to the relatively regular dispersion of SiO2-Tb3+ nanoparticles in PVP nanofibers, the rate of non-radiative transition (Anrad) decreases with the decrease of energy transfer between molecules.” This sentence confuse me because nonradiative transitions may exist within the same molecule. And even if they are nearby, energy transfer between them is not possible. Increase of the life time in my opinion is more related to the decrease of OH molecules on the surface due to presence there a polymer, and second is filling factor that is dependent on the index of refraction on the border molecule/polymer.
- I’m not sure what authors want to say by writing “However, the longer the fluorescence lifetime is, the greater the energy storage capacity is [22,23]”. This idea should be developed.
- Fig 6b should be presented only in ln scale, there is no need to show decay in linear scale. Y axis should be described as “Normalized Intensity”
- I think that one paper should be also cited in this manuscript: Nadège Francolon, Audrey Potdevin, Damien Boyera, Geneviève Chadeyron, Rachid, Mahiou, Luminescent PVP/SiO2@YAG:Tb3+ composite films, Ceramics International, 41, 9, 11272-11278, 2015,
Author Response
Dear professor,
Many thanks for your review, according to your suggestion, we tried our best to answer all the questions one by one and improve the manuscript
We appreciate for your kindly and helpful work, and hope that the correction will meet with approval.
Point 1: In the materials source of phenanthroline, acetylacetone, phenanthroline should be added
Response: We have added the following materials in Page 2 at line74 to 75 in the materials source.
1,10-Phenanthroline (Phen, 99%, AR), acetylacetone(Hacac, 98%, AR), hydrochloric acid (HCl, 38%, A.R.), hydrogen peroxide(H2O2, 30%, AR)
Point 2: Authors explain lower emission intensity at lower pH as “Under the condition of strong acid, the fluorescence intensity of Tb(acac)3phen drops sharply, due in part to the protonatation of the imines, which reduces the complexation with Tb3+”, but preparation method include strong acid (HCl) for all samples, and in the next steps only sodium hydroxide is added. It means that pH only increases and is alkaline. Therefore this explanation cannot be applied here
Response: Thank you for your careful work. Hydrochloric acid was used in the preparation of Terbium chloride hexahydrate, and the excess of hydrochloric acid was evaporated after crystallization. After that, ethanol was added to the crystallized powder to obtain a 0.1 mol/L TbCl3 solution, there is no strong acid (HCl) in TbCl3 solution.
Point 3: In the figure 3a instead of energy diagram it would be better to add integrated emission intensity in the function of pH, because from the picture it is not clear how the emission intensity changes. Excitation wavelength should be added in the Fig 3a,b
Response: We have added excitation wavelength in the Fig 3a,b, and added integrated emission intensity in the function of pH instead of energy diagram in the figure 3a.
Point 4: Authors wrote “by monitoring the strongest emission wavelength of the Tb3+ ions at 548 nm. These bands are attributed to the 5D4→7F6, 5D4→7F5, 5D4→7F4, and 5D4 →7F3 transitions of Tb3+ ions, respectively”, but emission at 548 is assigned only to , 5D4→7F5 transition
Response: For this question, we revised the manuscript and added the following sentences in Page 6 at Line 209 to 213.
As can be seen from Fig. 4b, the characteristic emission peaks of Tb3+ appear at 491 nm, 548 nm, 581 nm and 620 nm. These bands are attributed to the 5D4→7F6, 5D4→7F5, 5D4→7F4, and 5D4 →7F3 transitions of Tb3+ ions. The 5D4→7F5 transition at the emission wavelength of 548 nm is the hypersensitive transition with the strongest fluorescence.
Point 5: Authors wrote “The excitation energy emits the strongest fluorescence at 325 nm.” or “The emission peaks are assigned to the π-π* transition of the acetylacetone and phenanthroline” In the excitation spectra at 325 we don’t observe emission. It is energy that is absorbed to emit at 548 nm. Therefore this sentence is not correct.
Response: We change the sentence as following, in Page 6 at Line 206 to 207.
“The strongest emission fluorescence is obtained by excited at 325 nm. The excitation peaks are assigned to the π-π* transition of the acetylacetone and phenanthroline.”
And We have re-written this part to explain the excitation and emission spectra according to both of the reviewers’ suggestion in Page 6 at Line 192 to 213 as following.
The excitation spectrum for Tb(acac)3phen complex is very different from that of the SiO2-Tb3+ hybrid nanoparticles. In the excitation spectra of SiO2-Tb3+ nanospheres, the corresponding broadband absorption transition of organic ligand b-diketone still exists. The spectral line width is 25 nm, but the absorption band position of organic ligand remains unchanged, the excitation spectra of SiO2-Tb3+ nanospheres are more narrowed compared with pure Tb(acac)3phen complex, because intramolecular energy transfer occurs much more effectively between the ligands and Tb3+ following silica encapsulation. The excitation peaks are assigned to the π-π* transition of the acetylacetone and phenanthroline [17]. The absorption at 325 nm in the excitation spectrum of SiO2-Tb3+ is stronger than the corresponding absorption in the Tb(acac)3phen complex, indicating that SiO2 nanoparticles participate in the coordination of phenanthroline and acetylacetone molecules; an anhydrous preparation of SiO2-Tb3+ nanoparticles leads to different symmetry of the environment. Due to the effect of rigid silica shell, the nanoclusters of rare earth complexes can not be agglomerated. The specific surface area of rare earth complexes nanoclusters is larger, and there are more molecules on the surface than those occupying the interior of clusters. we consider that the increase of excitation intensity is due to the increase of molecules on the surface of rare earth complexes. The strongest fluorescence is obtained by excited at 325 nm, indicating that the fluorescence intensity of the complex is strongest under this excitation wavelength. Both the emission spectra of Tb(acac)3phen and SiO2-Tb3+ are obtained by monitoring the strongest emission wavelength of the Tb3+ ions at 548 nm. As can be seen from Fig. 4b, the characteristic emission peaks of Tb3+ appear at 491 nm, 548 nm, 581 nm and 620 nm. These bands are attributed to the 5D4→7F6, 5D4→7F5, 5D4→7F4, and 5D4 →7F3 transitions of Tb3+ ions. The 5D4→7F5 transition at the emission wavelength of 548 nm is the hypersensitive transition with strongest fluorescence intensity [17,18,19].
Point 6: In one sentence authors wrote “The SiO2-Tb3+/PVP hybrid nanofibers display bright green light upon radiation with ultraviolet light at 365 nm.” and in another “…the Tb3+ ion and is responsible for producing bright green light upon 325 nm ultraviolet radiation”. What is the real excitation wavelength used in the experiment. Another thing is that it should be also added in Fig 6.
Response: It is very sorry that we didn’t explain clearly. Ultraviolet light is a kind of UV light, it emits a wavelength of 365nm, we usually use this kind of light to check whether the sample can emit visible light or not under this kind of light, it is a kind of convenient method during experiment. But in fluorescence measurement, it is little different. Since we can see from the excitation spectrum in fig.4.a, the excitation spectrum of Tb(acac)3phen show a wide band, so we can choose a wavelength from 275nm to 400nm in measuring the fluorescent property, the 325nm is the best wavelength by which we can obtain the highest emission at 548nm. Therefore, we use 365 to explain the macroscopical phenomenon, while we use 325nm in measurement to obtain the strongest emission.
Point 7: I’m not sure what authors want to say by writing “However, the longer the fluorescence lifetime is, the greater the energy storage capacity is [22,23]”. This idea should be developed
Response: For this sentence, we think that the longer the decay time of exponential fluorescence have the more energy released when there is no energy transfer, and the more energy absorbed from ground state to excited state, the greater the energy storage capacity is.
Point 8: Authors wrote “At the same time, due to the relatively regular dispersion of SiO2-Tb3+ nanoparticles in PVP nanofibers, the rate of non-radiative transition (Anrad) decreases with the decrease of energy transfer between molecules.” This sentence confuse me because nonradiative transitions may exist within the same molecule. And even if they are nearby, energy transfer between them is not possible. Increase of the life time in my opinion is more related to the decrease of OH molecules on the surface due to presence there a polymer, and second is filling factor that is dependent on the index of refraction on the border molecule/polymer.
Response: We have re-written this part seriously by your suggestion as following in Page 8 at Line 267 to 278.
Non-radiative transitions between rare earths can transfer energy through phonons. Rare earth ions emit multiple phonons from the excited state at the same time, then relaxing to the next adjacent level. This process is called multiphonon non-radiative relaxation. The non-radiative decay of 5D4 of Tb3+ includes the non-radiative transition of 5D4-7Fj (j=3, 4, 5, 6) and the energy transfer from 5D4 to other Tb3+ ions. Since the relatively regular dispersion of SiO2-Tb3+ nanoparticles in PVP nanofibers, the energy shift caused by energy transfer of phonons can be neglected. The surface defect states (OH molecules) of SiO2 nanoparticles decrease sharply after SiO2-Tb3+ nanoparticles doped into PVP fibers, while OH-ions have larger vibration energy, they can greatly increase the multi-phonon non-radiative relaxation between energy levels, therefore the multi-phonon relaxation between energy levels decreases after the reduction of OH- ions, then the non-radiative transition probability decreased.
Point 9: Fig 6b should be presented only in ln scale, there is no need to show decay in linear scale. Y axis should be described as “Normalized Intensity”
Response: Thank you for your suggestion. We have revised Fig 6.b only in ln scale.
Point 10: think that one paper should be also cited in this manuscript: Nadège Francolon, Audrey Potdevin, Damien Boyera, Geneviève Chadeyron, Rachid, Mahiou, Luminescent PVP/SiO2@YAG:Tb3+ composite films, Ceramics International, 41, 9, 11272-11278, 2015,
Response: This is a great relevant published paper to our research, and it concludes a lot of useful theoretical explanations for experiments, it is helpful to us. It’s a pity we didn’t find it at the beginning, and now we added it in the revised version with No [18] in eferences.

Reviewer 2 Report
- Article presents a novel anhydrous route to preparing Tb encapsulated SiO2 nanoparticles. These nanoparticles were dispersed in a hybrid fibre for photonic applications. Processing, dispersion and properties of the above material were discussed.
- Article is well written and thoroughly written for most part.
- I recommend authors to present the particle size distribution.
Author Response
Dear professor,
Many thanks for your review, according to your suggestion, we tried our best to answer all the questions and improve the manuscript.
We appreciate for your kindly and helpful work, and hope that the correction will meet with approval.
Point 1:I recommend authors to present the particle size distribution
Response 1:According to your suggestion, we added the particle size distribution

Reviewer 3 Report
The authors reported the preparation of of SiO2 nanoparticles containing fluorescent Tb3+ complexes with acetyl acetone (acac) and phenanthroline (phen) (SiO2-Tb3+ 58 ) and of the Polyvinylpyrrolidone (PVP) ultrafine fibers containing SiO2-Tb3+ nanoparticles by electrospinning followed by a comparative study of their optical properties.
1. Preparation and optical properties of Tb(acac)3phen complex was reported before by Y Zheng Mat Letters 54 (2002), p.424 and L. Rino et al Journal of Non-Crystalline Solids 354 (2008) 5326–5327.
There are reports about the anhydrous synthesis of SiO2 spheres …is the preparation method novel or a variation?
The authors have to reformulate and improve the introduction by using other references for the material (see above and other) and other relevant in the field. The RE-ions starting paragraph is not useful.
2. Excitation and emission spectra are unclear and poorly explained…As “the encapsulation of Tb(acac)3phen complexes in silica nanoparticles does not change the crystal structure of the Tb(acac)3phen complex” I do see any reason for changes of the excitation spectra in Fig 4(a). Thabt UV band is more likely due to the Tb3+ intraconfigurational transition 4f-5d.
As the crystal structure of the Tb(acac)3phen complex SiO2-Tb3+ nanoparticles is preserved in the SiO2 I do not see how the PVP polymer enveloping can change the Tb3+ environment and effectively increasing the rate of radiation transition (Arad). In fact the luminescence lifetimes are very close (Fig. 6b). The discussion is not convincing…it needs to be reformulated.
3. What is shown in the inset of the Figure 6a? Why the spectra are slightly shifted? I think there is an experimental error.
4. How can fluorescence lifetimes can improve the luminescence monochromaticity?????!!!!
Other:
It is unclear the radiation attenuation in “…It can be seen from formula (1) that the greater the radiation attenuation…”
And “The excitation energy emits the strongest fluorescence…”
Author Response
Dear professor,
Many thanks for your review, according to your suggestions, we tried our best to answer all the questions one by one and improve the manuscript
We appreciate for your kindly and helpful work, and hope that the correction will meet with approval.
Point 1: Preparation and optical properties of Tb(acac)3phen complex was reported before by Y Zheng Mat Letters 54 (2002), p.424 and L. Rino et al Journal of Non-Crystalline Solids 354 (2008) 5326–5327.
There are reports about the anhydrous synthesis of SiO2 spheres …is the preparation method novel or a variation?
The authors have to reformulate and improve the introduction by using other references for the material (see above and other) and other relevant in the field. The RE-ions starting paragraph is not useful.
Response 1: In this study, we developed a variation method in preparation of anhydrous Silica-carried Terbium complex nanoparticles, it is based on the preparation of nano-SiO2. However, compared with previous preparation methods, the structure of nanoparticles were improved without water by lots of trying.
At the same time, we reformulate and improve the introduction according to your suggestion and add more references including the two mentioned in suggestions. They are great relevant published paper to our research, and they conclude a lot of useful theoretical explanations for experiments, it is helpful to us. It’s a pity we didn’t find it at the beginning, and now we added it in the revised version with No. [2]and No. [3] in references.
We revised introduction in Page 1 from Line 34 to 51 as following:
Studies on Rare-earth doped Polymer-based light emission materials are of great interest because of their have played a pivotal role in the fields of medical diagnostics, displays and detection systems [1]. Although the organic light emitters based on polymer materials show very good prospects, unfortunately, it has limited color purity and low luminous efficiency of those materials [2,3]. Therefore, finding materials with stable, narrow-linewidth and high luminous efficiency is the purpose of research. Rare-earth elements have a special electron configuration, imparting uncommon photonic properties. The luminescence characteristics are mainly based on the transitions of their 4f electrons within the f-f configuration or between the f-d configurations. Since the f-f transitions of rare-earth ions belong to forbidden transitions, the absorption cross section of rare earth ions in the visible or ultraviolet region is very small. However, organic ligands often have large absorption cross sections in the ultraviolet region, and energy can be transferred to rare-earth ions through intramolecular energy transfer, greatly improving the emission intensity of rare-earth ions [4-6]. We have selected terbium (Tb3+) elements with strong luminescent properties (green characteristic fluorescence) as a luminous center. In addition, the matching of energy levels between rare earth ions and organic ligands is one of the main factors affecting the luminescent efficiency of rare earth complexes, because of the energy levels of Terbium and acetylacetone (acac) are matched properly. In this section, acetylacetone and phenanthroline are used as ligands to synthesize Tb (acac)3Phen complexes
[2]Zheng Y.; Lin J.; Liang Y.; et al. A novel terbium (III) beta-diketonate complex as thin film for optical device application[J]. Mater. Lett. 2002, 54, 0-429.
[3]Rino L.; Simoes W.; Santos G .; et al. Photo and electroluminescence behavior of Tb(ACAC)3phen complex used as emissive layer on organic light emitting diodes[J]. J. Non-Cryst. Solids. 2008, 354, 5326-5327.
[18]Francolon, N.; Potdevin, A.; Boyera D.; Chadeyron, G.; Rachid.; Mahiou.; Luminescent PVP/SiO2@YAG:Tb3+ composite films. Ceram. Int. 2015, 41, 11272-11278.
Point 2: Excitation and emission spectra are unclear and poorly explained…As “the encapsulation of Tb(acac)3phen complexes in silica nanoparticles does not change the crystal structure of the Tb(acac)3phen complex” I do see any reason for changes of the excitation spectra in Fig 4(a). Thabt UV band is more likely due to the Tb3+ intraconfigurational transition 4f-5d.
Response 2: We added explanation for the excitation and emission spectra in the revised manuscript as following.
The excitation spectrum for Tb(acac)3phen complex is very different from that of the SiO2-Tb3+ hybrid nanoparticles. In the excitation spectra of SiO2-Tb3+ nanospheres, the corresponding broadband absorption transition of organic ligand ß-diketone still exists. The line width of the excitation spectrum is 25 nm, but the absorption band position of organic ligand remains unchanged, the excitation spectrum of SiO2-Tb3+ nanospheres are more narrowed compared with pure Tb(acac)3phen complex, because intramolecular energy transfer occurs much more effectively between the ligands and Tb3+ following silica encapsulation. The excitation peaks are assigned to the π-π* transition of the acetylacetone and phenanthroline [17]. The absorption at 325 nm in the excitation spectrum of SiO2-Tb3+ is stronger than the corresponding absorption in the Tb(acac)3phen complex, indicating that SiO2 nanoparticles participate in the coordination of phenanthroline and acetylacetone molecules; An anhydrous preparation of SiO2-Tb3+ nanoparticles leads to different symmetry of the environment. Due to the effect of rigid silica shell, the nanoclusters of rare earth complexes can not be agglomerated. The specific surface area of rare earth complexes nanoclusters is larger, and there are more molecules on the surface than those occupying the interior of clusters. we consider that the increase of excitation intensity is due to the increase of molecules on the surface of rare earth complexes. The strongest fluorescence is obtained by excited at 325 nm, indicating that the fluorescence intensity of the complex is strongest under this excitation wavelength. Both the emission spectra of Tb(acac)3phen and SiO2-Tb3+ are obtained by monitoring the strongest emission wavelength of the Tb3+ ions at 548 nm. As can be seen from Fig. 4b, the characteristic emission peaks of Tb3+ appear at 491 nm, 548 nm, 581 nm and 620 nm. These bands are attributed to the 5D4→7F6, 5D4→7F5, 5D4→7F4, and 5D4 →7F3 transitions of Tb3+ ions, The 5D4→7F5 transition at the emission wavelength of 548 nm is the hypersensitive transition with strongest fluorescence intensity [17,18,19]. The SiO2-Tb3+ hybrid nanoparticles have stronger luminescence than the corresponding Tb(acac)3phen pure complex; the electric-dipole transition and its corresponding intense emission at 548nm have significantly increased [19]. The amorphous silica nanoparticles provide a microenvironment for the terbium molecules that decrease nonradiative transitions by confinement resulting in the increasing of fluorescence intensity of SiO2-Tb3+.
As the crystal structure of the Tb(acac)3phen complex SiO2-Tb3+ nanoparticles is preserved in the SiO2 I do not see how the PVP polymer enveloping can change the Tb3+ environment and effectively increasing the rate of radiation transition (Arad). In fact the luminescence lifetimes are very close (Fig. 6b). The discussion is not convincing…it needs to be reformulated.
We have re-written this part according to the question.
TEM shows that SiO2-Tb3+ nanoparticles are enveloped by PVP polymer, which demonstrates that the site of Tb3+ ions is located in an environment without inversion symmetry, and effectively increasing the rate of radiation transition (Arad). The change of radiation transition rate dependent on the index of refraction on the border molecule/polymer. The increase of radiation transition rate to Tb3+ in SiO2-Tb3+/PVP hybrid fibers is due to the increase of refractive index of external medium. At the same time, due to the relatively regular dispersion of SiO2-Tb3+ nanoparticles in PVP nanofibers, the rate of non-radiative transition (Anrad) decreases with the decrease of energy transfer between molecules. Non-radiative transitions between rare earths can transfer energy through phonons. Rare earth ions emit multiple phonons from the excited state at the same time, then relaxing to the next adjacent level. This process is called multiphonon non-radiative relaxation. The non-radiative decay of 5D4 of Tb3+ includes the non-radiative transition of 5D4-7Fj (j = 3, 4, 5, 6) and the energy transfer from 5D4 to other Tb3+ ions. Since the relatively regular dispersion of SiO2-Tb3+ nanoparticles in PVP nanofibers, the energy shift caused by energy transfer of phonons can be neglected. The surface defect states (OH molecules) of SiO2 nanoparticles decrease sharply after SiO2-Tb3+ nanoparticles doped into PVP fibers, while OH- ions have larger vibration energy, they can greatly increase the multi-phonon non-radiative relaxation between energy levels, therefore the multi-phonon relaxation between energy levels decreases after the reduction of OH- ions, then the non-radiative transition probability decreased. Therefore, when the reduction of non-radiation rate (Anrad) is greater than the increase of radiation rate (Arad), the sum of the two factors will decrease.
Point 3: What is shown in the inset of the Figure 6a? Why the spectra are slightly shifted? I think there is an experimental error.
Response 3: The inset of the Fig. 6a is SiO2@Tb(acac)3phen/PVP spinning solution under 365 ultraviolet light, since we don’t have a optical fluorescent microscope to characterize fibers with a diameter of 120 nm, I would like to shown that our spinning solution can emit bright green light under UV light.
Emission spectra of SiO2-Tb3+ nanosphere was obtained by excited solid powder at 325 nm, and emission spectra of SiO2@Tb(acac)3phen/PVP obtained by excited spinning solution at 325nm. The normalized emission spectra for SiO2-Tb3+ nanosphere and SiO2@Tb(acac)3phen/PVP shown in Fig. 6 (a), we want to explain that SiO2-Tb3+/PVP hybrid fibers and SiO2-Tb3+ nanoparticles have similar emission peaks.
For the π-π* transition which is the main type of fluorescence emission, the excited state of electrons is more stable than the ground state. With the increase of solvent polarity, the excited state ratio has a greater stabilization effect on the ground state, so the energy of π-π* transition is reduced and the red shift occurs.
In fact, we combined the two spectra for comparation, if we show the emission spectra of (a)SiO2-Tb3+ hybrid nanoparticles and (b)SiO2@Tb(acac)3phen/PVP individually, the intensity is much different.
Point 4: How can fluorescence lifetimes can improve the luminescence monochromaticity?????!!!!
Response 4: We are very sorry for our unclear writing. SiO2@Tb(acac)3phen nanoparticles and SiO2@Tb(acac)3phen/PVP fibers both have long fluorescence lifetime and good color purity. It can be observed from Figure 6(a), emission spectra of SiO2-Tb3+ nanospheres and SiO2@Tb(acac)3phen/PVP fibers have narrow band at 5D4-7F5 energy level. The narrower the excitation linewidth is, the narrower the frequency range of the spectral line is, which proves the better monochromaticity of light and color purity. The linewidth is determined by the lifetime of the excited state of the luminescent substance and the relative motion state between the light source and the observer. The narrower the linewidth corresponding to the transition emission is., the smaller the dispersion of its energy, that is, the longer the average lifetime of the excited state atom is.

Round 2
Reviewer 1 Report
The manuscript can be published in the present form.
Author Response
Many thanks for your review
Reviewer 3 Report
The authors have improved the manuscript and reached the acceptance level for the publication....
The authors have to state clearly that the Tb-complexes were synthetised and studied before and their new contribution to the development of such materials.
Author Response
Point 1: The authors have to state clearly that the Tb-complexes were synthetised and studied before and their new contribution to the development of such materials
Response 1: We added some researches as following in Page 2 from Line 45 to 49 of Tb-complexes and their contribution to the development of this kind of materials.
L. Rino et al. successfully prepared Tb(acac)3phen complex and used as emissive layer on Organic Light Emitting Diodes (OLED); Youxuan Zheng developed Tb(acac)3AAP and worked on its photoluminescent, electroluminescent properties; Bukvetskii, B.V. studied the atomic structure of crystals of Tb(acac)3phen which characterized by means of the X-ray structural analysis method.
